# Expression of RBMS3 in Breast Cancer Progression

**DOI:** 10.3390/ijms24032866

**Published:** 2023-02-02

**Authors:** Tomasz Górnicki, Jakub Lambrinow, Monika Mrozowska, Hanna Romanowicz, Beata Smolarz, Aleksandra Piotrowska, Agnieszka Gomułkiewicz, Marzena Podhorska-Okołów, Piotr Dzięgiel, Jędrzej Grzegrzółka

**Affiliations:** 1Division of Histology and Embryology, Department of Human Morphology and Embryology, Wroclaw Medical University, 50-368 Wroclaw, Poland; 2Laboratory of Cancer Genetics, Department of Pathology, Polish Mother’s Memorial Hospital Research Institute, Rzgowska 281/289, 93-338 Lodz, Poland; 3Division of Ultrastructure Research, Department of Human Morphology and Embryology, Wroclaw Medical University, 50-368 Wroclaw, Poland

**Keywords:** RNA-binding protein 3 (RBMS3), carcinogenesis, cancer prevention, target discovery, target therapy, epithelial–mesenchymal transition (EMT)

## Abstract

The aim of the study was to evaluate the localization and intensity of RNA-binding motif single-stranded-interacting protein 3 (RBMS3) expression in clinical material using immunohistochemical (IHC) reactions in cases of ductal breast cancer (in vivo), and to determine the level of RBMS3 expression at both the protein and mRNA levels in breast cancer cell lines (in vitro). Moreover, the data obtained in the in vivo and in vitro studies were correlated with the clinicopathological profiles of the patients. Material for the IHC studies comprised 490 invasive ductal carcinoma (IDC) cases and 26 mastopathy tissues. Western blot and RT-qPCR were performed on four breast cancer cell lines (MCF-7, BT-474, SK-BR-3 and MDA-MB-231) and the HME1-hTERT (Me16C) normal immortalized breast epithelial cell line (control). The Kaplan–Meier plotter tool was employed to analyze the predictive value of overall survival of *RBMS3* expression at the mRNA level. Cytoplasmatic RBMS3 IHC expression was observed in breast cancer cells and stromal cells. The statistical analysis revealed a significantly decreased RBMS3 expression in the cancer specimens when compared with the mastopathy tissues (*p* < 0.001). An increased expression of RBMS3 was corelated with HER2(+) cancer specimens (*p* < 0.05) and ER(−) cancer specimens (*p* < 0.05). In addition, a statistically significant higher expression of RBMS3 was observed in cancer stromal cells in comparison to the control and cancer cells (*p* < 0.0001). The statistical analysis demonstrated a significantly higher expression of *RBMS3* mRNA in the SK-BR-3 cell line compared with all other cell lines (*p* < 0.05). A positive correlation was revealed between the expression of RBMS3, at both the mRNA and protein levels, and longer overall survival. The differences in the expression of RBMS3 in cancer cells (both in vivo and in vitro) and the stroma of breast cancer with regard to the molecular status of the tumor may indicate that RBMS3 could be a potential novel target for the development of personalized methods of treatment. RBMS3 can be an indicator of longer overall survival for potential use in breast cancer diagnostic process.

## 1. Introduction

According to the WHO GLOBOCAN 2020 report data, breast cancer (BC) is the most commonly diagnosed cancer, with nearly 2.3 million new cases worldwide in 2020 [1]. It is also the fifth leading cause of cancer mortality, responsible for 6.9% of all cancer-related deaths in 2020 [1]. Based on the Global Cancer Observatory forecast, by the year 2030, the number of new cases will increase to 2.7 million per year [2]. Genetic and molecular analyses have allowed researchers to identify four main intrinsic subtypes of BC: luminal A, luminal B, HER2-enriched and triple-negative breast cancer (TNBC, also called basal-like) [3]. As indicated in Table 1, each subtype differs in its expression of biomarkers, especially estrogen receptor (ER), progesterone receptor (PR), and human epidermal growth factor receptor 2 (HER2) [4,5]. The expression of these biomarkers plays a significant role, among other anatomical features, in estimating the prognosis of BC [3].

RBMS3 is a glycine-rich protein that belongs to the family of c-Myc gene single-strand binding proteins (MSSPs). RBMS3, similarly to other MSSPs, is involved in processes that are crucial for cell life, such as cell-cycle progression and apoptosis [6,7]. RBMS3 participates in various processes, both physiological and pathological, e.g., embryogenesis or liver fibrosis [8,9]. Published papers indicate that RBMS3 can be viewed as a regulating factor of carcinogenesis in various cancers, including ovarian and nasopharyngeal cancers [10,11,12]. RBMS3 is postulated to regulate the progression of nasopharyngeal cancer by influencing the expression of the p53 protein, becoming a potential regulator of the cell cycle in this type of cancer [12]. In ovarian cancer, it can be involved in drug-resistance mechanisms [10]. It has been reported that RBMS3 participates in the carcinogenesis process of breast cancer, since it is often described as a suppressor protein. In recent studies, authors have provided data that point to the fact that a certain level of RBMS3 is necessary for cancer progression [13,14,15,16,17,18]. The currently postulated mechanisms explaining the role of RBMS3 in the progression of breast cancer include involvement in the epithelial–mesenchymal transition (EMT) by inhibiting the Wnt/β-catenin signaling pathway and other EMT-related transcription factors, such as TWIST1 or PRRX1 [12,17,19,20]. Another mechanism of influence of RBMS3 in breast cancer is its presence in the miR-141-3p/RBMS3 axis that inhibits proliferation and promotes apoptosis in breast cancer cells [15]. Another study reported data related to RBMS3′s suppression leading to downregulation of cell programmed death ligand-1 (PD-L1) in TNBC, resulting in increased anti-tumor immune activities [18]. There is also evidence that the expression of RBMS3 in the stroma cells of breast cancer could have an impact on the progression of BC [16]. Although RBMS3 seems to play a major role in carcinogenesis, there remains a need for extensive research because of its complex influence on breast cancer.

The aim of this study is to discuss the role of RBMS3 in breast cancer. Using immunohistochemical staining performed on paraffin-embedded blocks of breast cancer samples and molecular analysis performed with breast cancer cells from cell-line cultures, we showed the correlation between RBMS3 levels and particular intrinsic subtypes of BC. A further aim of this study is to discuss RBMS3 as a novel potential therapeutic target and biomarker of overall survival in breast cancer.

## 2. Results

### 2.1. The Immunohistochemical Intensity of RBMS3′s Expression Varies in Cancer Cells, the Stroma of the Tumor, and the Control Mastopathy Cases, Exhibiting a Dependence on the Expression of Crucial Breast Cancer Receptors

The analysis of the immunohistochemical expression of RBMS3 in 490 cases of IDC and 26 cases of mastopathy showed a statistically significant decrease in RBMS3 expression in the cancer specimens compared to the mastopathy samples (Mann–Whitney test *p* < 0.001, Figure 1a, Figure 2a,b). Furthermore, the statistical analysis of the clinical data together with the immunohistochemical expression of RBMS3 showed a significantly increased expression of RBMS3 in the cancer cells of the HER2 positive cases (Mann–Whitney test *p* < 0.05, Figure 1b). Meanwhile, increased expression of RBMS3 correlated with the negative status of the estrogen receptor (Mann–Whitney test *p* < 0.05, Figure 1c). However, there was no statistically significant difference in the expression of RBMS3 in cancer cells between progesterone-positive and -negative cases.

Expression of RBMS3 in the stroma of the cancer cases was significantly higher than in the control specimens (Mann–Whitney test *p* < 0.0001, Figure 3a, Figure 2c,d). Moreover, RBMS3 expression in TNBC samples was significantly lower than in the other molecular types (Mann–Whitney test *p* < 0.001, Figure 3b). Further investigation in the stroma of breast cancer showed significant increases in RBMS3 expression in the specimens with positive expression of the progesterone receptor and samples with positive expression of the estrogen receptor (respectively, Mann–Whitney test *p* < 0.01 and *p* < 0.001, Figure 3c,d). On the other hand, we observed no correlation of RBMS3 expression with expression of the HER2 receptor. RBMS3 expression in the stroma of IDC was significantly higher than in the cancer cells (Mann–Whitney test *p* < 0.0001, Figure 3e).

There were no statistically significant differences in the expression of RBMS3 with regard to the grade, TNM, and stage of the cancer. This absence of difference was observed in the cancer cells and the stroma.

### 2.2. In Vitro Analysis of RBMS3 Expression Differs from RBMS3 Expression in Clinical Material

For further investigation of the difference in RBMS3 expression in the different molecular types of breast cancer, we performed an RT-qPCR analysis of *RBMS3* expression at the mRNA level in the chosen cell lines representing the various molecular types of breast cancer. When compared to the control Me16C cell line (ANOVA and Bonferroni’s multiple comparison test *p* < 0.05, Figure 4a), the expression of *RBMS3* was significantly different (mostly lower) in all the examined cell lines, with the only exception being the SK-BR-3 cell line. *RBMS3* expression was highest in the SK-BR-3 cell line among all the investigated cell lines. The Western blot analysis of the protein expression showed a significantly higher expression of RBMS3 in the control cell line than in the MCF-7 and BT-474 cancer cell lines (ANOVA and Bonferroni’s multiple comparison test *p* < 0.05, Figure 4b) There is a visible and statistically significant trend that the more aggressive types of breast cancer, including TNBC and HER-2-positive cancers, presented higher expression of RBMS3 than their benign counterparts.

### 2.3. RBMS3 Expression May Be an Indicator of Longer Overall Survival

The analysis of the clinical data regarding the survival of patients showed shorter overall survival in the group of patients without an IHC expression of RBMS3 (Gehan–Breslow–Wilcoxon test *p* = 0.051, Figure 5). The univariate and multivariate Cox analyses of the overall survival indicated that only G, pT, and pN were independent prognostic factors (Table 2).

Additionally, using the Kaplan–Meier estimator we performed an analysis of the *RBMS3* mRNA expression of 2976 cases of breast cancer. This revealed that the group of patients with lower *RBMS3* expression (cut-off point: median) had statistically significant shorter overall survival (*p* < 0.0001, Figure 6) [21].

## 3. Discussion

RBMS3 is reported to be deregulated in many different types of neoplastic processes, for example, gastric cancer, esophageal squamous cell carcinoma, breast cancer, or gall bladder carcinoma [13,22,23,24]. In this study, we have discussed the role of RBMS3 in the progression of breast cancer with particular emphasis on receptor expression and the molecular type. We provided an analysis of RBMS3 expression in clinical material and cell lines, and presented experimental data supporting the statement of the potential role of RBMS3 expression in tumor stromal cells.

The results of our experiments apparently support previous studies’ results that indicate the downregulation of RBMS3 expression in breast cancer cells [13,19] and its correlation with negative estrogen-receptor status [14]. In addition, we discovered another potential interaction of RBMS3 with a positive HER2-receptor status, supported by an immunostaining analysis and the high expression of *RBMS3* at the mRNA level in the SK-BR-3 line (representing the HER2-enriched subtype). At the protein level, the expression levels of RBMS3 in the SK-BR-3 and MDA-MB-231 cell lines were the highest among all the examined breast cancer cell lines and were significantly higher than in MCF-7 and BT-474 cell lines. A significantly higher expression of RBMS3 in more aggressive types of tumors characterized by the lack of estrogen-receptor expression, and in the case of the SK-BR-3 cell line the presence of the HER2 receptor, may indicate that a certain level of RBMS3 expression is necessary for specific types of cancer progression and their ability to create metastasis [20,25]. RBMS3′s anticancer function could be related to the mechanisms that regulate adhesiveness and invasiveness, which are also associated with the EMT process in cancer. These findings are in partial agreement with recent reports that provide evidence of RBMS3 knockdown resulting in the impairment of in vivo tumor growth and a decreased level of angiogenesis [17,18]. It is important to mention that the research carried out by Block et.al and Zhu et.al was conducted only on the triple-negative type of breast cancer cells. The results provided in this study support the claim that RBMS3 expression in the TNBC and HER-2-enriched types is similarly high as in the control cell lines, meaning that RBMS3 could possibly act as a suppressor in Lum-A and Lum-B types of breast cancer. They may also suggest that a normal level of RBMS3 expression is necessary for the growth of TNBC and HER-2-enriched types of breast cancer; this observation requires more detailed investigation.

The role of the tumor microenvironment (TME) is a topic of rapidly increasing interest among scientists [26]. The TME is the unique environment in which the tumor develops. It consists of an extracellular matrix, blood vessels, signaling molecules, and multiple types of cells that play a pivotal role in tumor cancerogenesis by stimulating and facilitating uncontrolled cell proliferation [27,28]. Stromal cells are an integral part of the TME. Alongside other elements, they play a part in the maintenance of cancer stemness by promoting angiogenesis, invasion, metastasis, and chronic inflammation [29]. The transcriptomic analysis of the *RBMS3* gene’s expression in the stromal cells of breast cancer provides evidence of its being gradually downregulated through all three grades of breast cancer [16]. The results that we present suggest a higher expression of the RBMS3 protein in the stroma cells of breast cancer compared with the mastopathy control cases or the cancer cells, with no significant differences between grades. Together, these data suggest that RBMS3′s deregulation in the stroma of the tumor may influence the role of stromal cells in breast cancer through currently unknown mechanisms. Furthermore, there may exist a currently unknown post-transcriptional mechanism regulating the expression of RBMS3 in the stroma of the tumor, which could explain the grade-dependent expression of *RBMS3* and the lack of grade dependency at the protein level. A negative correlation of RBMS3 expression in the stromal cells with TNBC, and a positive one with ER- and PR-receptor status of the tumor, may indicate that there is a possibility for RBMS3 to display an antitumor effect depending on the molecular characteristics of the tumor. Specifically, a negative correlation with TNBC may indicate the tumor-suppressor role of RBMS3 in breast cancer stroma. A higher expression of RBMS3 in the stroma of breast cancer may indicate the potentially important role of the TME in the progression of IDC.

In addition to its potential antitumor properties, the expression of RBMS3 may be an indicator of overall survival. These capabilities were reported by scientists researching lung squamous cell carcinoma and gastric cancer [23,24,30]. In this current study, we provide evidence of RBMS3′s potential use as a positive prognostic marker of overall survival in breast cancer. The results of our clinical data analysis are consistent with the findings of Wang et al. [14]. The analysis of *RBMS3* mRNA expression in samples from the GEO and EGA data repositories also supports the suggestion that RBMS3 may be a useful tool for breast cancer diagnosis. The analysis of RBMS3 expression can be included as a supplementary category in defining prognosis of patient survival based on the molecular characteristics of the tumor, increasing the accuracy of predictions. The correlation of RBMS3 expression with TNBC and the expression of progesterone receptor may also lead to the distinction of new molecular subtypes of breast cancer based on the analysis of combined biomarkers.

Taking into consideration all the results presented in this study, we provide evidence of a potential novel explanation of RBMS3′s role in breast cancer. Currently available reports have tried to explain RBMS3′s anticancer activity in all types of breast cancer through the inhibition of the Wnt/β-catenin pathway and the inhibition of the epithelial–mesenchymal transition process (EMT), mainly by impacting on TWIST, PRRX1, or MMP2 [14,17,19]. Our results suggest that further studies should be conducted to consider the differences in RBMS3 expression correlated with receptor expression in cancer cells and stromal cells. We distinguished a positive correlation with overall survival, supporting the idea of a potential tumor-suppressing role for the expression of RBMS3 in breast cancer stroma. These findings open the way for further studies to unveil the exact role and mechanisms of these correlations.

Although further studies on the exact molecular mechanisms underlying the role of RBMS3 in breast cancer are required, RBMS3 may be potentially used in the development of novel therapeutic and diagnostic approaches in breast cancer. These may target not only cancer cells but also tumor stroma cells, making these therapies more complex and potentially adaptive to the patient’s type of tumor, which would translate into a more personalized approach to patient treatment.

## 4. Materials and Methods

### 4.1. Patients’ Characteristics

The clinical material consisted of 524 paraffin blocks with clinical data from patients operated on for IDC. The clinical and pathological characteristics of the patients are presented in Table 3. Additionally, 26 paraffin blocks and clinical data from cases of mastopathy were analyzed as a control for the breast cancer cases. Patients’ clinical material was obtained from the Division of Pathomorphology of the Polish Mother’s Memorial Hospital Research Institute. The experiment was performed in accordance with the ethical standards and following the approval of the Ethics Committee of Wroclaw Medical University (decision no. KB 625/2022 25.08.2022).

### 4.2. Tissue Microarrays (TMAs)

A total of 21 TMAs were prepared from 524 cases of IDC and 26 cases of mastopathy. Prior to performing TMA blocks, the histological slides stained with hematoxylin and eosin were obtained from whole samples of breast cancer and mastopathy cases stored in the form of paraffin blocks (donor blocks). The slides were scanned using the Pannoramic Midi II histological scanner (3DHISTECH Ltd, Budapest, Hungary). After that, using the Pannoramic Viewer program 1.15.4 (3DHISTECH Ltd.), representative areas from the entire sections where selected. In addition, to increase the representativeness of each case, 3 representative cores each with a size of 1.5 mm were selected from the donor blocks and transferred to the TMA ‘recipient’ block using the TMA Grand Master system 2.6.6.69657 (3DHISTECH Ltd.).

### 4.3. Immunohistochemistry

The paraffin blocks with the breast cancer and mastopathy cases were cut into 4-µm sections. The immunohistochemical reactions were performed using anti-RBMS3 rabbit polyclonal antibody (Catalog # PA5-57028, Invitrogen, Thermo Fisher Scientific, Waltham, MA, USA) in a 1:200 dilution. The immunohistochemical reactions were performed using a Dako Autostainer Link 48 (Dako, Glostrup, Denmark). The visualization of the reactions was carried out using EnVision™ FLEX High pH (Link, Glostrup, Denmark) reagents (Dako), according to the manufacturer’s instructions. The IHC reactions for 490 cases of IDC were suitable for the further analysis. The IHC reaction for RBMS3 antigen was assessed using the immunoreactive scale (IRS) by Remmele and Stegner [31], that evaluates the percentage of positive cancer cells (A) and the intensity of color reaction (B). The final score is the product of the values A and B (see Table 4).

### 4.4. Kaplan–Meier Plotter

The Kaplan–Meier plotter tool was used for correlation of *RBMS3* mRNA expression with overall survival [21]. This is a tool for Kaplan–Meier plot generation based on data from GEO, EGA, and TCGA. *RBMS3* mRNA expression data was split into two groups for analysis: “high expression” and “low expression” using the median as the cut-off value.

### 4.5. Cell Lines

Four breast cancer cell lines were used in the experiments, representing types of tumors of increasing aggressiveness (MCF-7: luminal A, BT-474: luminal B, SK-BR-3: HER2-enriched, and MDA-MB-231: triple-negative), along with a normal cell line: immortalized breast epithelial cell line (HME1-hTERT) (Me16C). All cell lines were provided by ATCC (American Type Culture Collection ATCC^®^, Old Town Manassas, VA, USA). Respective culture media were used to provide optimal conditions for cell growth: MEBM (Lonza, Basel, Switzerland) for the Me16C cell line, αMEM (Lonza) for the MCF-7 and BT-474 cell lines, McCoy’s (ATCC) for the SK-BR-3 cell line, L-15 (Lonza) for the MDA-MB-231 cell line. All media contained 1% l-glutamine and penicillin-streptomycin solution, as well as 10% fetal bovine serum (Sigma-Aldrich^®^, St. Louis, MO, USA). The cells were passaged with the use of TrypLE™ (Gibco, Thermo Fisher Scientific, Waltham, MA, USA) when they were at approximately 70% confluence.

### 4.6. Real-Time PCR

Real-time PCR was applied to determine the relative level of *RBMS3* mRNA expression in the analyzed cell lines (MDA-MB-231, SK-BR-3, BT-474, MCF-7, Me16C). Total RNA was isolated with the use of a RNeasy mini kit (Qiagen, Hilden, Germany), according to the manufacturer’s instructions. Reverse transcription reactions were performed with the use of iScript™ cDNA synthesis kit (Bio-Rad, Hercules, CA, USA). The conditions of the reactions were as follows: priming for 5 min at 25 °C, reverse transcription for 20 min at 46 °C, and final inactivation of reverse transcriptase for 1 min at 95 °C. RT-qPCR was carried out in 20-µL volumes using the TaqMan Universal PCR MasterMix (Applied Biosystems, Foster City, CA, USA). The reactions were performed using a 7500 Real-time PCR system and iTaq™ Universal Probes Supermix (Bio-Rad), according to the manufacturer’s instructions. The TaqMan probes employed were Hs01104892_m1 for RBMS3 (Applied Biosystems) and endogenous control gene Hs99999903_m1 for β-actin (Applied Biosystems), further used for normalization purposes. The experiments were run in triplicate. The reactions were carried out under the following conditions: initial denaturation for 2 min at 94 °C, followed by 40 cycles of denaturation (15 s, 94 °C) and annealing with elongation (1 min, 60 °C). The relative *RBMS3* mRNA expression levels were calculated using the ΔΔCt method.

### 4.7. Western Blotting

Whole cell lysates were obtained from the BC cell lines (MDA-MB-231, SK-BR-3, BT-474, and MCF-7) and the control cell line (Me16C) using CelLytic™ MT Cell Lysis Reagent (Sigma-Aldrich) with the addition of Halt™ Protease Inhibitor Cocktail 100x (Thermo Fisher Scientific) and 2 mM PMSF (phenylmethylsulphonyl fluoride) (Sigma-Aldrich). The protein level was determined through colorimetric analysis with the use of bicinchoninic acid (Pierce BCA Protein Assay Kit) and NanoDrop 1000 (Thermo Scientific). The lysates were mixed with 4X SDS-PAGE gel-loading buffer (200 mM Tris-HCl—pH 6.8, 400 mM DTT, 8% SDS, 0.4% bromophenol blue, 40% glycerol) for 10 min at 95 °C, loaded onto 10% acrylamide gel and separated by SDS-PAGE under reducing conditions, then transferred onto a PVDF membrane in the XCell SureLock™ Mini gel electrophoresis system (Thermo Fisher Scientific). After the protein transfer, the membrane was incubated in a blocker solution (4% BSA in TBST buffer) for 1 h at RT, followed by overnight incubation at 4 °C with anti-RBMS3 monoclonal rabbit antibody, (Catalog # PA5-57028, Invitrogen, Thermo Fisher Scientific). Subsequently, the membrane was washed with TBST with 0.1% Tween-20 and incubated for 1 h at RT with secondary antibody (Jackson ImmunoResearch, Mill Valley, CA, USA) diluted at 1:3000, then rinsed and treated with Luminata Classico (Merck KGaA, Darmstadt, Germany) chemiluminescent substrate. Rabbit anti-human β-actin monoclonal antibody (#4970; Cell Signaling Technology, Danvers, MA, USA) diluted 1:1000 was used as an internal control. The Western blotting results were analyzed using the ChemiDoc MP system (Bio-Rad). The experiments were run in triplicate.

### 4.8. Statistical Analysis

The Kolmogorov–Smirnov test was applied to evaluate the normality assumption of the groups examined. The Mann–Whitney and ANOVA with Bonferroni’s multiple comparison post hoc tests were conducted to compare the differences in the expression of the examined markers in all groups of patients in vitro and in the clinicopathological data. Additionally, the Spearman’s correlation test was applied to analyze the existing correlations. The Kaplan–Meier method was used to construct survival curves. The Gehan–Breslow–Wilcoxon method was applied and univariate and multivariate Cox analyses of survival were performed to evaluate the survival analysis. All statistical analyses were conducted using Prism 9.0 (GraphPad Software) and Statistica 13.3 (Tibco Software, Inc.). The results were considered statistically significant when *p* < 0.05.

## Figures and Tables

**Figure 1 ijms-24-02866-f001:**
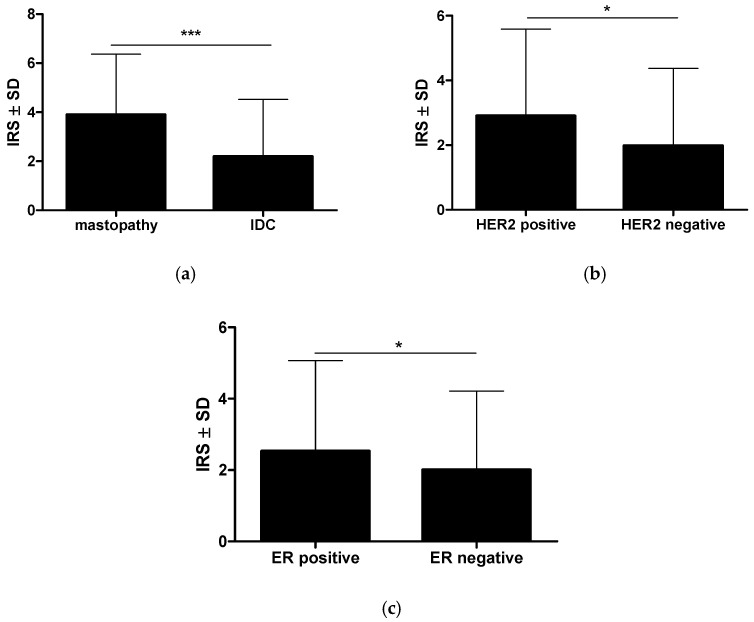
(**a**) The statistical analysis revealed a significantly higher RBMS3 expression as assessed by the immunoreactive score in the control mastopathy cases compared with the cancer cells in breast cancer. (**b**) The breast cancer cells with positive expression of HER2 and (**c**) estrogen receptors presented a higher expression of RBMS3 compared with tumors lacking expression of these receptors (Mann–Whitney test * *p* < 0.05; *** *p* < 0.001) (IDC—invasive ductal carcinoma, IRS—immunoreactive score).

**Figure 2 ijms-24-02866-f002:**
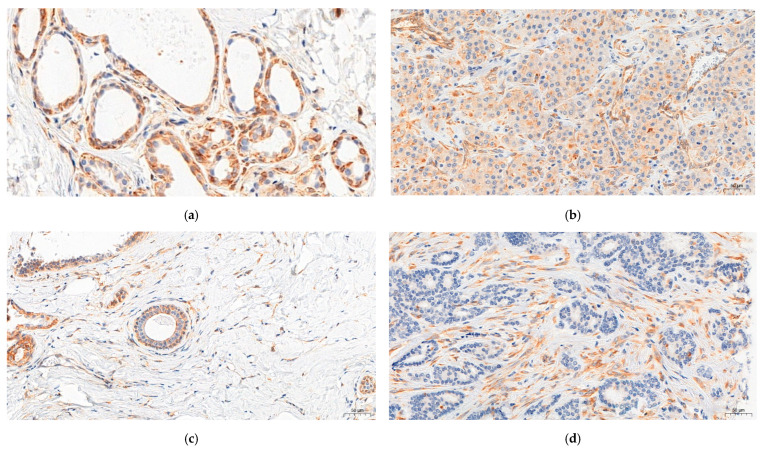
Immunohistochemical visualization of RBMS3 expression. (**a**) High cytoplasmic expression of RBMS3 in mastopathy cases. (**b**) Low expression of RBMS3 in breast cancer cells. (**c**) Low cytoplasmic expression of RBMS3 in the stroma of mastopathy cases and (**d**) high expression in the stroma cells of the breast cancer cases. Magnification ×200.

**Figure 3 ijms-24-02866-f003:**
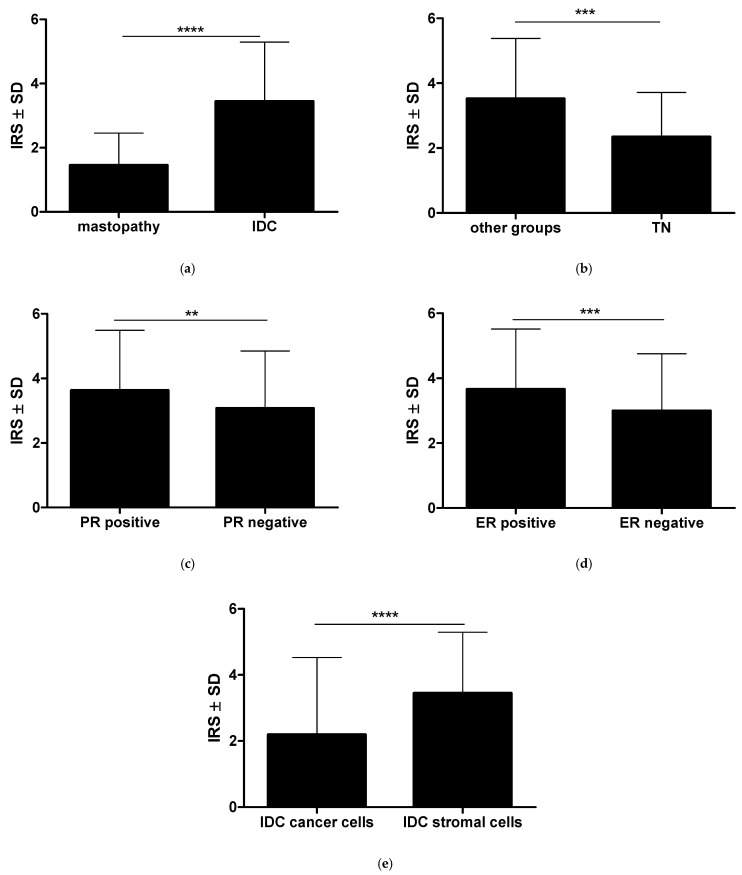
Analysis of RBMS3 expression in the stroma of breast cancer: (**a**) RBMS3 expression in the stroma of breast cancer was statistically higher than in the mastopathy cases; (**b**) Triple-negative (TN) cases of breast cancer displayed lower expression of RBMS3 in the stroma than the other molecular types of breast cancer combined. (**c**) Significant increase in RBMS3 expression in the specimens with positive expression of the progesterone receptor and (**d**) with positive expression of the estrogen receptor. The expression of RBMS3 was statistically lower in the cancer cells than in the stromal cells of the breast cancer specimens (**e**). (Mann–Whitney test ** *p* < 0.01, *** *p* < 0.001, **** *p* < 0.0001) (IDC—invasive ductal carcinoma).

**Figure 4 ijms-24-02866-f004:**
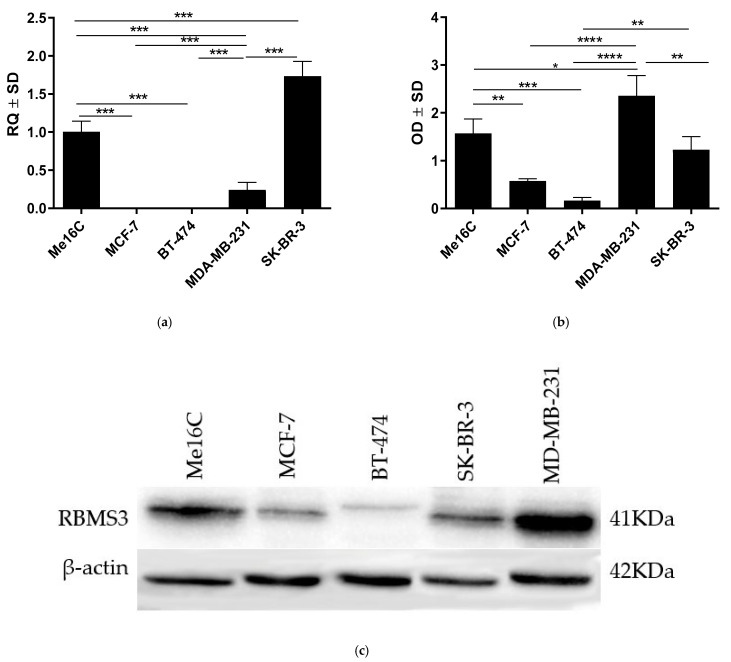
In vitro analysis of RBMS3 expression in breast cancer cell lines representing the four main molecular types of breast cancer and a control cell line (Me16C). (**a**) The statistical analysis of *RBMS3*′s expression at the mRNA level showed a significantly different expression of *RBMS3* in all the examined cell lines in comparison to the control Me16C cell line. (**b**,**c**) Analysis at the protein level showed a significantly higher expression of RBMS3 in the MDA-MB-231 and SK-BR-3 cell lines than in the MCF-7 and BT-474 cell lines; ((**a**) Bonferroni’s multiple comparison test, (**b**) Bonferroni’s multiple comparison test, * *p* < 0.05, ** *p* < 0.01, *** *p* < 0.001, **** *p* < 0.0001).

**Figure 5 ijms-24-02866-f005:**
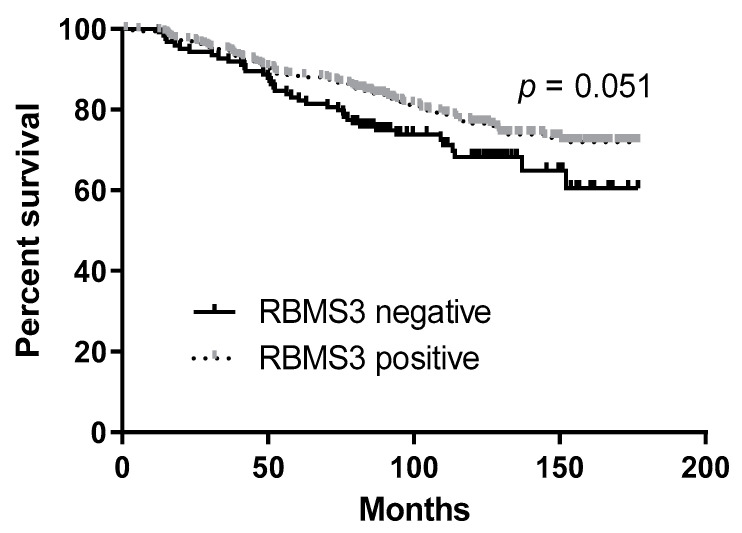
The analysis of the clinical data regarding the survival of patients showed shorter overall survival in the group of patients without IHC expression of RBMS3 (Gehan–Breslow–Wilcoxon test *p* = 0.051).

**Figure 6 ijms-24-02866-f006:**
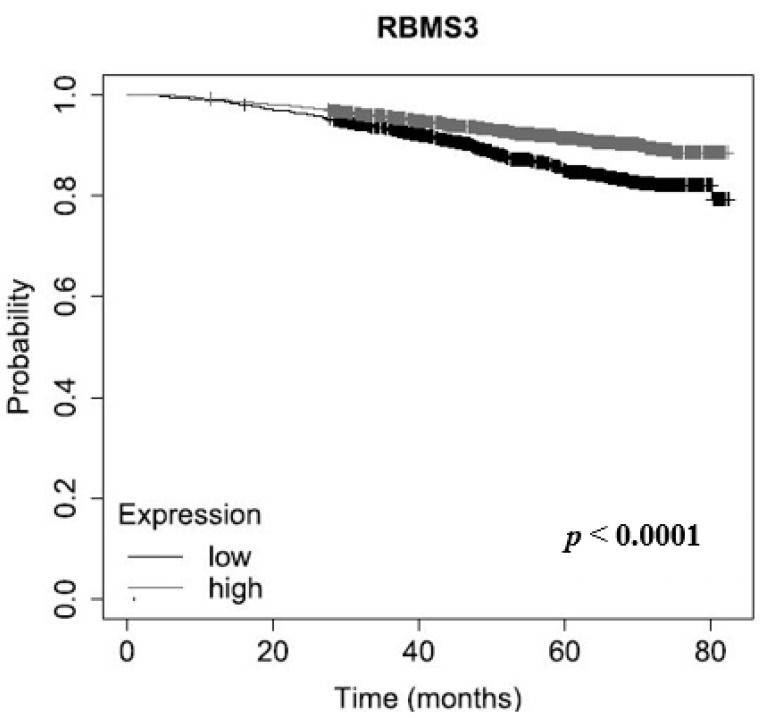
Analysis of *RBMS3* mRNA expression in 2976 cases of breast cancer, using the Kaplan–Meier estimator. The analysis revealed a significant positive correlation between the expression of *RBMS3* and overall survival (*p* < 0.0001) [21].

**Table 1 ijms-24-02866-t001:** Different molecular subtypes of breast cancer. Each molecular subtype is defined by the expression of three main receptors: estrogen receptor, progesterone receptor, and human epidermal growth factor receptor 2 [4].

Molecular Subtype of Breast Cancer	Receptor Status		
	Estrogen Receptor (ER)	Progesterone Receptor (PR)	Human Epidermal Growth Factor Receptor 2 (HER2)
Luminal A	+	≥20%	-
Luminal B	+	<20%	+/-
HER2-enriched	-	-	+
Triple-negative breast cancer (TNBC)	-	-	-

**Table 2 ijms-24-02866-t002:** Univariate and multivariate Cox analyses of overall survival in cases of invasive ductal carcinoma.

	Univariate Cox Analysis of Survival				Multivariate Cox Analysis of Survival			
Characteristics	*p*-Value	Hazard Ratio	HR 95% CI Lower	HR 95% CI Upper	*p*-Value	Hazard Ratio	HR 95% CI Lower	HR 95% CI Upper
G1 vs. G2-G3	**<0.0100**	**3.0873**	**1.5179**	**6.2792**	**<0.0100**	**2.5309**	**1.2509**	**5.1208**
pT1 vs. pT2-pT4	**<0.0001**	**2.4469**	**1.7123**	**3.4966**	**<0.0010**	**2.0371**	**1.4201**	**2.9221**
pN0 vs. pN1-pN3	**<0.0001**	**2.6544**	**1.8541**	**3.8001**	**<0.0001**	**2.1583**	**1.4997**	**3.1062**
ER negative vs. ER positive	0.2260	0.7987	0.5550	1.1493				
PR negative vs. PR positive	0.1416	0.7626	0.5313	1.0946				
HER2 0-HER2 2 vs. HER2 3	0.4485	1.3206	0.64338	2.7105				
Triple-negative vs. other groups	0.3742	1.3843	0.67566	2.8361				
RBMS3 IRS stromal: 0 vs. 1–12	0.3196	0.6548	0.2844	1.5075				
RBMS3 IRS cancer: 0 vs. 1–12	**<0.0500**	**0.6470**	**0.4470**	**0.9365**	0.1429	0.7576	0.5226	1.0983

ER—estrogen receptor, PR—progesterone receptor, RBMS3—RNA-binding motif single-stranded-interacting protein 3, IRS—immunoreactive scale, HR—hazard ratio, CI—confidential interval.

**Table 3 ijms-24-02866-t003:** Clinical and pathological characteristics of studied patients.

Parameters	Patients	
	IHC *n* = 524	%
**Age**		
≤60	165	31.49
>60	359	68.51
**Tumor grade**		
G1	87	16.60
G2	342	65.27
G3	92	17.56
No data	3	0.57
**Tumor size**		
pT1	325	62.02
pT2	168	32.06
pT3	3	0.57
pT4	9	1.72
No data	19	3.63
**Lymph nodes**		
pN0	314	59.92
pN1-pN3	180	34.35
pNx	30	5.73
**Stage**		
I	224	42.75
II	257	49.05
III	18	3.44
IV	0	0.00
**ER**		
Neg.	177	33.78
Pos.	344	65.65
No data	3	0.57
**PR**		
Neg.	183	34.92
Pos.	338	64.50
No data	3	0.57
**HER2**		
Neg.	272	51.91
Pos.	36	6.87
No data	216	41.22
**Molecular tumor types**		
Triple-negative	34	6.49
Other types	487	92.94
No data	3	0.57

**Table 4 ijms-24-02866-t004:** Graphic presentation of Remmele and Stegner scale showing the available values. The final score is the multiplication of A and B values (A × B) [31].

Points	Percentage of Positive Cancer Cells (A)	Intensity of Color Reaction (B)
0	0%	No color reaction
1	<10%	Mild reaction
2	10–50%	Moderate reaction
3	51–80%	Strong reaction
4	81–100%	

## Data Availability

The data presented in this study are available in this article and Appendix A attached to it.

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
