# Peer review of "Expression of RBMS3 in Breast Cancer Progression"

_ijms, 2023, doi:10.3390/ijms24032866_

Round 1

Reviewer 1 Report (Previous Reviewer 2)

I have already reviewed this MS once.

My major concern was the Western blot data.The beta actin blots are not professional.(I have checked the supplementary file)

Produce a high quality professional data of WB 

Thats my comment.

Author Response

In accordance with the comments and suggestions received from reviewers, our manuscript (ijms-1856638) has been revised.

Detailed responses to Reviewer’s comments are listed below:

            My major concern was the Western blot data. The beta actin blots are not professional.(I have checked the supplementary file). Produce a high quality professional data of WB 

Response: Thank you very much for your evaluation and recommendation. We conducted Western Blot experiments again, according to advises given of the reviewer,  in order to provide highest quality of analyzed data. New Western Blot data are presented in attached supplementary files. New repetitions of Western Blot was produced from the same cell material that the first set of data. Densitometric analysis of new blots revealed similar results across all 6 repetitions of RBMS3 and significant differences in beta actin samples. Statistical analysis of the new data led to reevaluation of RBMS3 expression in breast cancer cell lines changing the values of expression. New results are presented in the manuscript. The analysis of new data support our claims in greater extent than previous data. We are very grateful for finding and pointing out this crucial mistake that we had a chance to correct.

Reviewer 2 Report (Previous Reviewer 1)

Explanations to the previous comments are now included in the manuscript.

Research article can be published, some minor spell checks are required.

Author Response

In accordance with the comments and suggestions received from reviewers, our manuscript (ijms-1856638) has been revised.

Detailed responses to Reviewer’s comments are listed below:

Explanations to the previous comments are now included in the manuscript. Research article can be published, some minor spell checks are required.

Response: Thank you very much for your evaluation and recommendation. We reviewed manuscript and corrected language mistakes.  One of the reviewers advised us to repeat Western Blot experiments.

We created a new set of WB data from the same samples as the first one according to the recommendation of the reviewer. Significant changes were observed only in beta actin replicates. Analysis of new data led to changes in graphical presentation of RBMS3 expression in breast cancer cell lines. Statistical analysis of the data still supports all the previous claims of the manuscript adding additional statistically significant proof of RBMS3 expression being higher in more aggressive types of cancer. Original images of new Western Blots are attached as supplementary materials. We sincerely apologize for the mistake.

Reviewer 3 Report (New Reviewer)

This manuscript is dedicated to the actual problem of molecular oncology - searching and characterization of a new potential marker of breast cancer. However, the text is on the page 9: "The analysis of RBMS3 mRNA expression of the samples coming from the GEO and EGA data 234 repositories also support the statement that RBMS3 may be a useful tool in the breast cancer diagnostic process." It would be useful to briefly discuss in the Discussion section how RBMS3 expression is better/worse or can complement existing clinical and laboratory prognostic criteria of breast cancer.

Author Response

In accordance with the comments and suggestions received from reviewers, our manuscript (ijms-1856638) has been revised.

Detailed responses to Reviewer’s comments are listed below:

                This manuscript is dedicated to the actual problem of molecular oncology - searching and characterization of a new potential marker of breast cancer. However, the text is on the page 9: "The analysis of RBMS3 mRNA expression of the samples coming from the GEO and EGA data 234 repositories also support the statement that RBMS3 may be a useful tool in the breast cancer diagnostic process." It would be useful to briefly discuss in the Discussion section how RBMS3 expression is better/worse or can complement existing clinical and laboratory prognostic criteria of breast cancer.

Response: Thank you very much for your evaluation and recommendation. We added in the Discussion section additional explanation about potential of RBMS3 usage in diagnostic of breast cancer. “The analysis of RBMS3 expression can be used as a supplementary category in defining prognosis of patient survival based on the molecular characteristics of the tumor increasing accuracy of the predictions. The correlation of RBMS3’s expression with TNBC and the expression of progesterone receptor may also lead to the distinction of a new molecular subtypes of breast cancer based on the analysis of many combined biomarkers.”

One of the reviewers advised us to repeat Western Blot experiments. We created a new set of WB data from the same samples as the first one according to the recommendation of the reviewer.  Significant changes were observed only in beta actin replicates. Analysis of new data led to changes in graphical presentation of RBMS3 expression in breast cancer cell line. Statistical analysis of the data still supports all the previous claims of the manuscript adding additional statistically significant proof of RBMS3 expression being higher in more aggressive types of cancer. Original images of new Western Blots are attached as supplementary materials. We sincerely apologize for the mistake.

Round 2

Reviewer 1 Report (Previous Reviewer 2)

The authors have satisfactorily addressed my comments, and I may recommend the manuscript for publication

This manuscript is a resubmission of an earlier submission. The following is a list of the peer review reports and author responses from that submission.

Round 1

Reviewer 1 Report

Research article is not upto the mark to justify the title of the study.

There are several details which are missing in introduction about the role of RBMS3 in cancer progression. Introduction needs a lot of improvement.

No explanations are provided about the result of the experiment and what is the significance of doing the experiment. Experiments are not properly conducted with essential control like normal epithelial cells.

Attaching PDF of the paper with some suggestions.

Reviewer 2 Report

The MS “The expression of RBMS3 protein in invasive ductal breast cancer progressionpreliminary study” tried to explain the role of RBMS3 in Breast cancer progression.

Unfortunately, I am not happy with the way the manuscript is written and the data representation.

-Manuscript needed scientific editing.

The abstract and Introduction are not very clear to the readers.

-Western blot data-I didn’t understand anything from this Western blot data

-(it’s not professional)

Blots must be shown in the main figures.

The quality of the western blot is poor and there is no marker shown in the figures.

Where are the Beta actin controls for all the replicates?

-What is the molecular weight of RBMS3

-Also produce all raw data (IHC, RT PCR) as supplementary figures.